# Ranolazine Attenuates Brain Inflammation in a Rat Model of Type 2 Diabetes

**DOI:** 10.3390/ijms232416160

**Published:** 2022-12-18

**Authors:** Velia Cassano, Martina Tallarico, Giuseppe Armentaro, Caterina De Sarro, Michelangelo Iannone, Antonio Leo, Rita Citraro, Emilio Russo, Giovambattista De Sarro, Marta Letizia Hribal, Angela Sciacqua

**Affiliations:** 1Department of Medical and Surgical Sciences, Magna Græcia University, 88100 Catanzaro, Italy; 2Science of Health Department, Magna Græcia University, 88100 Catanzaro, Italy; 3System and Applied Pharmacology@University Magna Grecia, Science of Health Department, School of Medicine, Magna Graecia University of Catanzaro, 88100 Catanzaro, Italy; 4CNR-Institute for Biomedical Research and Innovation, 88100 Catanzaro, Italy

**Keywords:** Alzheimer’s, neurodegeneration, ranolazine, type 2 diabetes mellitus

## Abstract

Recent studies suggest a pathogenetic association between metabolic disturbances, including type 2 diabetes (T2DM), and cognitive decline and indicate that T2DM may represent a risk factor for Alzheimer’s disease (AD). There are a number of experimental studies presenting evidence that ranolazine, an antianginal drug, acts as a neuroprotective drug. The aim of the present study was to evaluate the effects of ranolazine on hippocampal neurodegeneration and astrocytes activation in a T2DM rat model. Diabetes was induced by a high fat diet (HFD) and streptozotocin (STZ) injection. Animals were divided into the following groups: HFD/STZ + Ranolazine, HFD/STZ + Metformin, HFD/STZ + Vehicle, NCD + Vehicle, NCD + Ranolazine and NCD + Metformin. The presence of neurodegeneration was evaluated in the hippocampal cornus ammonis 1 (CA1) region by cresyl violet staining histological methods, while astrocyte activation was assessed by western blot analysis. Staining with cresyl violet highlighted a decrease in neuronal density and cell volume in the hippocampal CA1 area in diabetic HFD/STZ + Vehicle rats, while ranolazine and metformin both improved T2DM-induced neuronal loss and neuronal damage. Moreover, there was an increased expression of GFAP in the HFD/STZ + Vehicle group compared to the treated diabetic groups. In conclusion, in the present study, we obtained additional evidence supporting the potential use of ranolazine to counteract T2DM-associated cognitive decline.

## 1. Introduction

While aging represents the strongest risk factor for Alzheimer’s dementia (AD), emerging studies have provided evidence for a significant association between metabolic disturbances associated with type 2 diabetes (T2DM), particularly insulin resistance, and cognitive decline; with a one- to five-fold increase in the incidence of AD in patients with T2DM compared to subjects without the disease [1].

Dysfunction in glucose metabolism produces a decrease in hippocampal volume and changes in microvasculature, which influence memory consolidation, learning ability and neurotransmission [2]. These observations led to the identification of a novel form of diabetes that involves the brain and has biochemical and molecular characteristics that overlap those of T2DM, such as an impaired function of several components of the insulin signalling pathway and gene coding for insulin-like growth factor 1 (IGF1) and 2 (IGF2) and their receptors [3], and has been named type 3 diabetes (T3DM) [1].

A study conducted on C57BL/6 mice demonstrated that T2DM/obesity produces histopathological, molecular and biochemical brain abnormalities typical of AD degeneration. In that study, mice were fed with a high fat diet for 16 weeks, their body weight doubled, they developed T2DM and their brain weight was significantly reduced. Those effects were linked to increased levels of IGF-1 receptor, insulin receptor substrate-1 (IRS-1) and -4 (IRS-4) and glial fibrillary acidic protein (GFAP), a marker of astroglial activation and astrocytosis, two AD-related conditions [4].

In a subsequent experimental study, rats were treated with a single intracerebral injection of streptozotocin (ic-STZ) to demonstrate that diabetes mellitus-type biochemical and molecular abnormalities could be produced in central nervous system (CNS) neurons and the brain by exposure to STZ. Animals were subjected to tests to assess spatial learning and memory, and their brains were analysed for indices of AD neurodegeneration. Results showed that ic-STZ treatments reduced oxidative metabolism and cerebral glucose utilization and inhibited insulin receptor function, causing an impairment in learning, cognitive behaviour and memory [1].

The data obtained in animal models were confirmed in human studies; structural brain changes related to aging, in particular atrophy and small vessel disease, were positive in diabetic subjects [5]. There is evidence that alterations in brain glucose homeostasis are intrinsic to the pathogenesis of AD and may arise several years before the onset of clinical manifestations [6]. In fact, in patients with AD, the brain regions vulnerable to amyloid accumulation show significantly higher tissue glucose concentrations, and even higher concentrations of glucose in the brain tissue are associated with greater severity of both amyloid plaque deposition and neurofibrillar disease [6,7]. In accordance with this, chronic hyperglycaemia can lead to abnormalities in the cerebral capillaries with impaired cerebral blood flow. An additional potential factor contributing to cognitive impairment in subjects affected by T2DM is brain microvascular damage.

Magnetic resonance imaging studies have shown that the risk of developing lacunae and hippocampal atrophy in subjects with T2DM was moderately increased, and the severity of the picture increased with the progression and duration of T2DM [8,9].

Therefore, the increased risk of cognitive impairment in subjects with T2DM could be attributable to the chronic stimulus of hyperglycaemia, peripheral insulin resistance, oxidative stress and microvascular damage [10]. In fact, neuronal cell death due to oxidative stress by advanced glycation end products (AGE) was increased in patients with AD and T2DM compared to those with AD or T2DM alone [11,12].

Ranolazine, *N*-(2,6-dimethylphenyl)-4(2-hydroxy-3-[2-meth-oxyphenoxy]-propyl)-1-piperazine acetamide dihydrochloride, is an active piperazine, approved by the Food and Drug Administration (FDA) in 2006 for the treatment of chronic angina pectoris and as an anti-arrhythmic [13]. In addition to its anti-ischemic and antianginal effects, ranolazine also showed the ability to lower HbA1c in patients with coronary artery disease and T2DM in two clinical studies [14].

Afterwards, several studies have evaluated ranolazine’s effect on the CNS, under the hypothesis that it could act as a neuroprotective drug [15]. It is known that ranolazine reduces cellular excitability of dorsal root ganglion neurons, suggesting that its effects may have applications in the treatment of CNS disorders. Accordingly, Kahlig et al. tested the hypothesis that ranolazine was able to suppress increased persistent current evoked by NaV1.1 mutant channels as a therapeutic strategy for SCN1A-associated epilepsy and migraine syndromes [16]. Ranolazine showed a sixteen-fold and five-fold higher inhibition of persistent current compared to tonic block and use-dependent block of peak current, respectively. However, ranolazine did not have important effects on current density, activation and voltage dependence of inactivation. The results from this study suggested a possible use of ranolazine for selective repression of enhanced persistent current as a potential new therapeutic strategy for familial neurological disorders connected with certain sodium channel mutations.

A study carried out on primary cultures of rat astrocytes and neurons investigated the effect of ranolazine on apoptosis, cell viability, inflammation and oxidative stress [17]. In cultured astrocytes, ranolazine significantly increased cell viability and proliferation at any tested concentration and reduced LDH loss, Smac/Diablo expression and Caspase 3 activity, indicating a lower rate of cell death. Ranolazine also increased the expression of the anti-inflammatory protein, PPAR-γ, and reduced the levels of pro-inflammatory proteins, IL-1 β and TNFα. Furthermore, the antioxidant proteins, Cu/Zn-SOD and Mn-SOD, increased significantly after the addition of Rn in cultured astrocytes.

Accordingly with previous studies, Elkholi and collaborators explored the neuroprotective effect of ranolazine versus pioglitazone in rats with diabetic neuropathy [18]. Treatment with ranolazine or pioglitazone for 6 weeks improved thermal hyperalgesia and allodynia. Moreover, results showed that ranolazine reduced TNF-α and IL-1β levels, in agreement with previous data, and upregulated PPAR-γ in cultured astrocytes. These data led to the hypothesis that ranolazine could act as a neuroprotective drug in the CNS by promoting astrocyte viability, preventing necrosis and apoptosis [18].

Recently, we explored the ability of ranolazine to counteract cognitive impairment and depressive-like behaviour and confirmed its hypoglycaemic effects in rats with a T2DM-like disease [19]. The aim of the present study was to focus on the CNS, characterising the effects of ranolazine on hippocampal neurodegeneration and astrocyte activation.

## 2. Results

### 2.1. Histology

In our previous study, we demonstrated that the glucose response during IPGTT was significantly improved in HFD/STZ + Ranolazine and HFD/STZ + Metformin groups compared to the HFD/STZ + Vehicle group. Furthermore, treatment with ranolazine or metformin significantly ameliorated learning and memory performances in treated groups compared to the diabetic untreated group. The effect of ranolazine was superimposable with that of metformin [19].

To investigate whether ranolazine reverses cognitive impairment in T2DM rats, cresyl violet staining was carried out to determinate the impact of 8 weeks of treatment on neurological death.

The hippocampus includes Cornus Ammonis, CA1 and CA2, consisting of a small pyramidal cell areas and CA3 and CA4 consisting of large pyramidal cell areas. The histological analysis performed for this work was focused on the CA1 area of the hippocampus.

Staining with cresyl violet (Nissl staining) highlighted a decrease in neuronal density and a decrease in cell volume in the hippocampal CA1 area in rats of the HFD/STZ + Vehicle group. In the NCD + Vehicle group, compared to HFD/STZ + Vehicle group, neuronal cells appeared as large conical cells with well delimited cytoplasm and round vesicular nuclei with evident nucleoli. The 8-week ranolazine treatment reduced T2DM-induced neuronal loss and neuronal damage in the HFD/STZ + Ranolazine group. A similar situation was observed in the HFD/STZ + Metformin group, with decreased neuronal loss compared to the HFD/STZ + Vehicle group. No differences were observed in the NCD + Ranolazine and NCD + Metformin groups compared to the NCD +Vehicle group (Figure 1). The neuronal cell count carried out in the hippocampal CA1 region showed that there are statistically significant differences between the six groups regarding neuronal viability, indicating a significant decrease in neuronal cells in the HFD/STZ + Vehicle group.

Direct comparison between relevant groups performed by Sidak’s test revealed no statistically significant differences between HFD/STZ + Ranolazine and HFD/STZ+ Metformin vs. the control groups, such as NCD + Vehicle, NCD + Ranolazine and NCD + Metformin, proving that treatment with ranolazine or metformin prevents neuronal death (Figure 2). In contrast, statistically significant differences were observed in the comparison between HFD/STZ + Vehicle vs. HFD/STZ + Ranolazine (*p* = 0.011), HFD/STZ + Vehicle vs. HFD/STZ + Metformin (*p* = 0.039) and HFD/STZ + Vehicle vs. NCD + Vehicle (*p* < 0.0001).

### 2.2. GFAP Expression

The inflammatory response in the brain is characterised by activation of glial cells and astrocytes, and the expression of GFAP is considered as a marker for this activation. In the present study, the expression of GFAP in the whole brain was quantified by western blot analysis. GFAP expression was significantly higher in the HFD/STZ + Vehicle group compared to the HFD/STZ + Metformin (1.8 ± 0.5 vs. 1.1 ± 0.2, Δ = +63.6%, *p* = 0.04) and NCD + Vehicle (1.8 ± 0.5 vs. 1.0 ± 0.2, Δ = +80%, *p* = 0.046) groups. Ranolazine treatment was able to reduce GFAP expression in comparison to untreated diabetic rats, but without a statistically significant difference (1.2 ± 0.3 vs. 1.8 ± 0.5, Δ = −33.3%, *p* > 0.05) (Figure 3).

### 2.3. Tumor Necrosis Factor α (TNF α) mRNA Levels

To further validate the anti-inflammatory properties of ranolazine, we assessed the mRNA levels of the master regulator of inflammatory activation [20], TNF-α, by real-time RT-PCR. We observed significantly higher TNF-α mRNA levels in HFD/STZ + Vehicle rats when compared to non-diabetic control groups (*p* = 0.01); notably, both metformin and ranolazine were able to significantly reduce TNF-α levels in diabetic rats (*p* = 0.01) (Figure 4).

## 3. Discussion

The current study was conducted to confirm that ranolazine is able to improve neuronal cell survival and the inflammatory profile in a rat model of T2DM. Metformin was used as a positive control as it is the first-line drug for hypoglycaemic treatment, in addition to also having positive effects on the prevention of cognitive decline in diabetic subjects [21].

Cognitive impairment is one of the major complications of T2DM; in fact, hyperglycaemia leads to a deficit in memory and learning, making T2DM patients vulnerable to developing AD [2].

Building on our previous data showing that ranolazine is able to counteract hyperglycemia-induced cognitive decline [19], in the present study, we confirmed the neuroprotective effect of ranolazine by the histological analysis of the CA1 region of the hippocampus. We observed that neuronal density was significantly lower in the HFD/STZ + Vehicle group compared to the HFD/STZ + Ranolazine group, highlighting the protective effect of ranolazine. The preventive effect, rather than an ameliorative effect, of ranolazine is supported by its ability to reduce neurodegeneration of hippocampal neurons after induction of T2DM. Moreover, our results are in agreement with previous studies, which demonstrated the neuroprotection and improvement in cognitive performance exercised by metformin [21].

To further validate the protective effect of ranolazine on the CNS, we quantified the protein levels of GFAP, a marker of astrocytes activation, whose levels are dysregulated in AD [22,23], and mRNA expression of the master regulator of inflammation, TNF-α, in biopsies [20] of brain tissue. Our results confirmed that ranolazine, as well as metformin, attenuated the damaging effects of hyperglycaemia.

In conclusion, the results described in this study confirmed the ability of ranolazine to counteract the impact of T2DM on cognitive function. The originality of this study was to analyse the effects of ranolazine on tissue, focussing on brain tissue, and then analysing astrocyte activation in a diabetic rat model. Ranolazine demonstrated consistent effects against neurodegeneration, as evidenced by the results obtained from histological analyses. These data support our, and others’, previous findings [19], consolidating the evidence that may suggest considering ranolazine as the drug of choice in individuals with T2DM with increased risk of cognitive decline. However, further studies are needed to elucidate the molecular mechanisms involved and to assess whether the effects of ranolazine are maintained in both sexes, as only male rats were analysed in the present study.

## 4. Material and Methods

### 4.1. Animals

Male 6-week-old Wistar rats (*n* = 48) were used in this study. Wistar rats were housed 2/3 per cage and kept under stable environmental conditions of humidity (60 ± 5%) and temperature (21 ± 2 °C), in a room with a 12/12-h reversed light/dark cycle (lights on at 20:00). Procedures involving animals and their care were performed in agreement with international and national law and policies (EU Directive 2010/63/EU for animal experiments, ARRIVE guidelines and the Basel declaration, including the B3R^ concept). The experimental protocols and the methods described herein were approved by the Animal Care Committee of the University Magna Graecia of Catanzaro. All efforts were made to minimise animal suffering and to reduce the number of animals used.

Diabetes was induced by switching 6-week-old rats to high fat diet (HFD: 59% fat, 15% protein, 20% carbohydrates), obtained from Laboratorio Piccioni S.R.L. Gessate Italy, for 50 days, and by administering two low doses of STZ (35 mg/kg bw i.p) at day 21 and day 42 from the beginning of dietary manipulation (HFD, *n* = 24). The control group (NCD, *n* = 24) were fed with a normal caloric diet (NCD) and injected with sodium citrate buffer. Diabetes induction was verified 8 days after the second dose of STZ by an intraperitoneal glucose tolerance test (IPGTT). Rats were deemed diabetic if blood glucose levels were >250 mg/dL at the end of the IPGTT (120 min after glucose load) [19].

Subsequently, diabetic rats were randomly divided into 3 groups (*n* = 8 for group) according to treatment: HFD/STZ + Vehicle (DMSO), HFD/STZ + Ranolazine or HFD/STZ + Metformin. Rats from the NCD control group were also randomly assigned to the NCD + Vehicle (DMSO), NCD + Ranolazine and NCD + Metformin groups (*n* = 8 for group). Treatments were carried out for eight consecutive weeks. Metformin was used as a positive control (Figure 5).

### 4.2. Tissue Preparation

Rats were anaesthetised using tiletamine/zolazepam 50 mg/kg i.p. (1:1; Zoletil 100; VIRBAC S.r.l., Milan Italy) to collect blood from the ophthalmic plexuses, and killed by transcardiac perfusion with cold phosphate buffer saline (PBS), pH 7.4, and subsequently with cold 4% paraformaldehyde (PFA) containing 0.2% saturated picric acid in phosphate buffered saline.

### 4.3. Histological Examination

Hippocampal cornu amonis 1 (CA1) region was analysed, in six animals per group, with cresyl violet to evaluate neuronal damage. Brains, fixed in 4% PFA, were quickly washed in bidistilled water (BD) and then cut to obtain the macroscopic block containing the area of interest (thickness about 5 mm, hippocampal area) and further washed in BD water. Subsequently, the samples were dehydrated using EtOH 50% (30 min), EtOH 70% (1 h), EtOH 96% (1 h), EtOH 100% (1 h) and organic reagent (histolemon) for 2 h. When the samples became diaphanous, they were included in liquid paraffin at 60 °C for 2 h and then left to solidify at 4 ° C. Paraffin embedded sections have been cut in the coronal plane at a thickness of 10 µm by a microtome and mounted on microscope slides. Paraffin-embedded brain sections were de-paraffined with xylene (20 min) and rehydrated with EtOH 100% (20 min), EtOH 80% (10 min), EtOH 70% (10 min), EtOH 50% (10 min) and distilled H_2_O (30 min). The cresyl violet dye was acidified with 1% acetic acid the slides were immersed and left in contact with the staining solution for 2.5 min.

The mean number of morphologically intact neurons per 100 µm length was counted in the CA1 hippocampal area to precisely estimate the extent of neuronal damage. Cell counting was performed in six serial sections per animal using a light microscope equipped with a 25x objective, as previously described [24].

### 4.4. Western Blotting for GFAP Expression

Frozen brain tissue samples were homogenised in a lysis buffer containing 1.5% Triton-X and protease inhibitors. Lysates were analysed by western blot, according to previously established protocols [25]. Proteins resolved by SDS-PAGE were electrophoretically transferred to nitrocellulose membranes. Non-specific binding sites on membranes were blocked by 5% non-fat dry milk in TBS/Tween20 (TBS-T) for 2 h. The membranes were incubated with primary antibody GFAP (1:500) overnight at 4 °C (GFAP (GA5) Mouse, Cell Signaling, Danvers, MA, USA). After washing with TBS-T, the membranes were incubated with anti-mouse peroxidase-conjugated secondary antibody (1:2500). Densitometric analysis was performed using ImageJ software (NIH, Bethesda, MD, USA). Equal protein loading was confirmed by reblotting the membranes with monoclonal antibody against β-actin (Sigma-Aldrich; Milan, Italy).

### 4.5. Real-Time RT-PCR for TNFα Expression

Frozen brain tissue samples were homogenised using Trizol reagent (Thermofisher Scientific, Waltham, MA, USA). Equal amounts of RNA were reverse transcribed and analysed by the quantitative real-time PCR technique using commercial Assays-on-Demand kits (all Applied Biosystems, Waltham, MA, USA) according to the manufacturer’s protocol and normalised to housekeeping 18S gene. All reactions were performed in duplicate. Relative gene expression was calculated by using the comparative CT method.

### 4.6. Statistical Analysis

Data are expressed as mean ± standard deviation (SD) or mean ± standard error (SE). The ANOVA test was used to test the differences between groups, and Sidak’s test was applied to compare data set pairs. A value of *p* ≤ 0.05 was considered significant. Comparisons were performed using SPSS 20.0 statistical software (SPPS, Inc., Chicago, IL, USA).

## Figures and Tables

**Figure 1 ijms-23-16160-f001:**
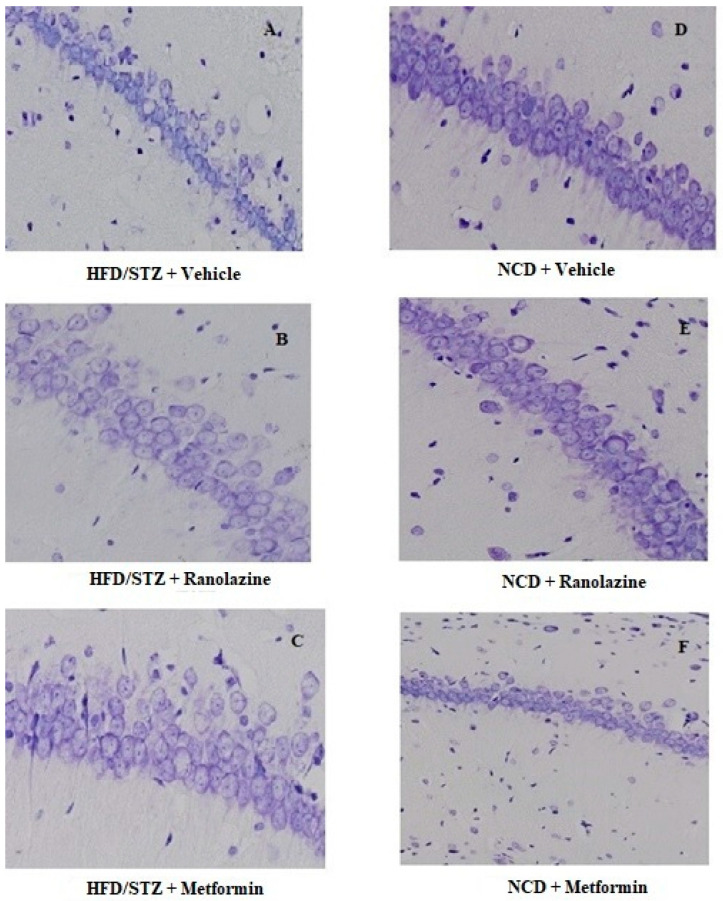
Representative images of histological analysis. Magnification 20x. Cresyl violet staining was performed on a section from the hippocampal CA1 region in (**A**) HFD/STZ + Vehicle group, (**B**) HFD/STZ + Ranolazine group, (**C**) HFD/STZ + Metformin group, (**D**) NCD + Vehicle group, (**E**) NCD + Ranolazine group, (**F**) NCD + Metformin group. HFD = high fat diet; NCD = normocaloric diet; STZ = streptozotocin.

**Figure 2 ijms-23-16160-f002:**
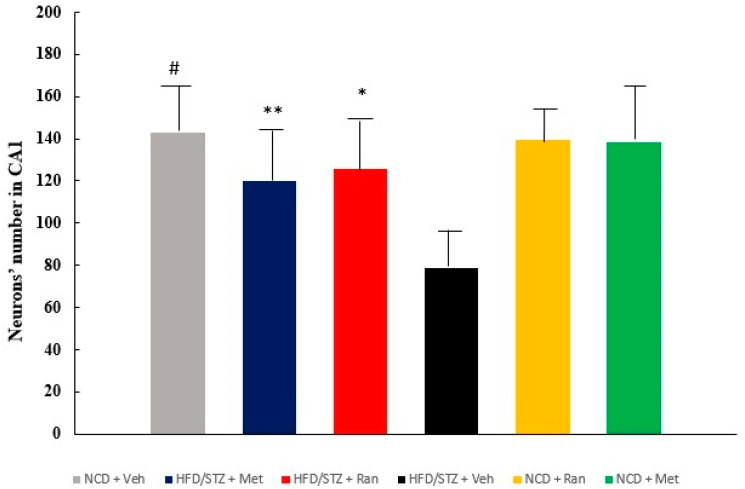
Neuronal count in the hippocampal CA1 region of the six study groups. Values are the mean ± SD (*n* = 6 per group). * *p* = 0.0011 HFD/STZ + Vehicle vs. HFD/STZ + Ranolazine, ** *p* = 0.039 HFD/STZ + Vehicle vs. HFD/STZ + Metformin, # *p* < 0.0001 HFD/STZ + Vehicle vs. NCD + Vehicle (Sidak’s test). HFD = high fat diet; NCD = normocaloric diet; STZ = streptozotocin.

**Figure 3 ijms-23-16160-f003:**
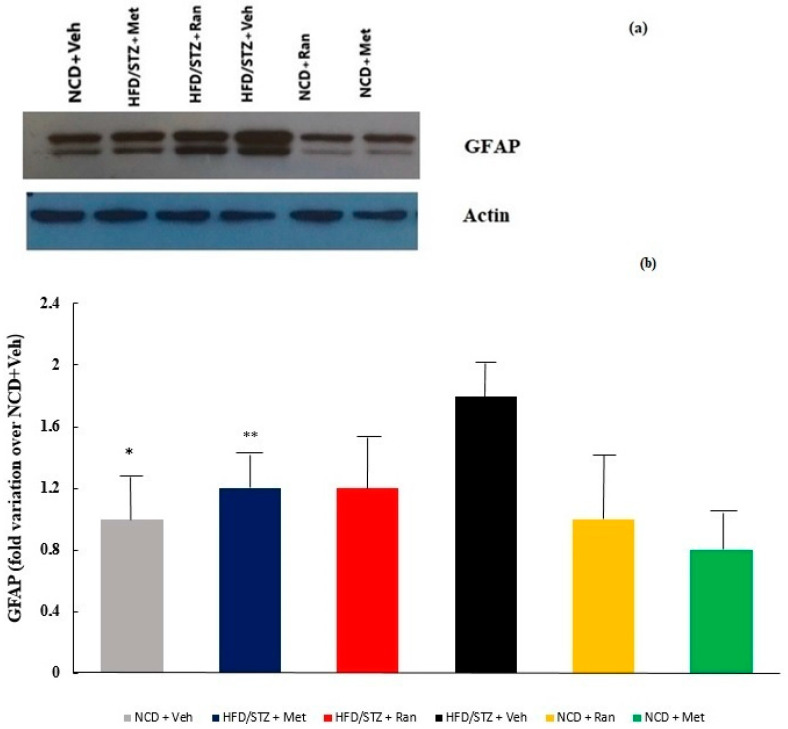
(**a**) Representative western blot images of glial fibrillary acidic protein (GFAP) (upper panel) and β-actin (lower panel). (**b**) Graph of GFAP variation in each experimental group. Data values are the mean ± SE (standard error) after normalization for β-actin, and expressed as fold variation over NCD + Vehicle group. (*n* = 3–4 per group). HFD = high fat diet; NCD = normocaloric diet; STZ = streptozotocin; Ran = ranolazine; Met = metformin; GFAP= glial fibrillary acidic protein. * *p* = 0.046 NCD + Vehicle vs. HFD/STZ + Vehicle, ** *p* = 0.04 HFD/STZ + Metformin vs. HFD/STZ + Vehicle.

**Figure 4 ijms-23-16160-f004:**
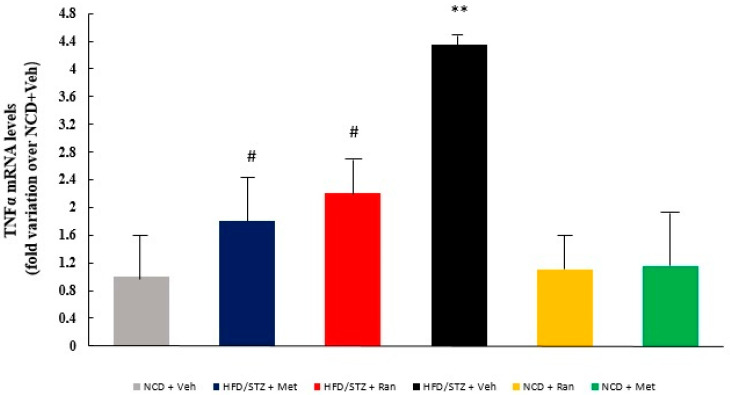
TNFα in each experimental group. Data values are the mean ± SE (standard error) after normalization for 18S levels, and expressed as fold variation over NCD + Vehicle group. (*n* = 3–4 per group.) HFD = high fat diet; NCD = normocaloric diet; STZ = streptozotocin; Ran = ranolazine; Met = metformin; GFAP= glial fibrillary acidic protein. ** *p* = 0.01 HFD/STZ + Vehicle vs. NCD + Vehicle, # *p*= 0.01 HFD/STZ + Metformin and HFD/STZ + Ranolazine vs. HFD/STZ + Vehicle.

**Figure 5 ijms-23-16160-f005:**
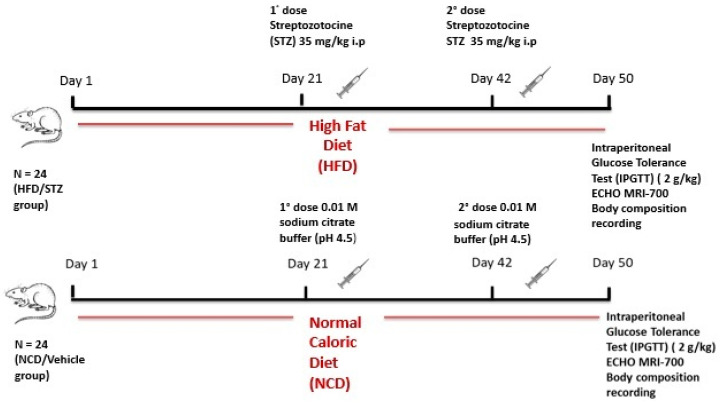
Schematic diagram of the experimental protocol.

## Data Availability

The data presented in this study are available in the article and Appendix A.

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
