# Peer review of "Ranolazine Attenuates Brain Inflammation in a Rat Model of Type 2 Diabetes"

_ijms, 2022, doi:10.3390/ijms232416160_

Round 1

Reviewer 1 Report

In the manuscript entitled “Ranolazine attenuates brain inflammation in a rat model of type 2 diabetes”, the authors set out to provide a possible mechanism for ranolazine to inhibit cognitive decline. The manuscript has been clearly presented and the English language is adequate. The introduction has been well thought through and well presented. 

Though the authors set out to decipher the mechanism involved in the inhibition of cognitive decline by ranolazine, the histological and GFAP expression data presented only reaffirms the role of ranolazine in the inhibition of cognitive decline and offer no mechanism for the inhibition. The manuscript needs to be projected as proof for the role of ranolazine in inhibition of cognitive decline rather than asserting as a possible mechanism for the inhibitory role.

There are overlaps between the introduction and discussion sections. The authors need to focus more on the results and discuss their findings. As a whole, the discussion section needs major improvement to provide value to the readers. 

Author Response

We thank the Reviewer for her/his appreciation of our manuscript and for the thoughtful comments that have helped us to improve it. We have rephrased the study aim according to the Reviewer suggestion and have extensively revised the discussion section to render it more focused, avoiding overlaps with the Introduction.

Reviewer 2 Report

The submitted manuscript describes an interesting, if limited, study investigating the protective effect of ranolazine on hippocampal inflammation in high-fat diet- and streptozotocin-induced diabetic rats. The major drawback to the study is that the major finding is based on quantitative western blot analysis. Western blots are only semi-quantitative and so other supportive data is necessary. For example, C-reactive protein levels, or Nrf2 expression and phosphorylation, as well as the protein expression and enzymatic activity of γ-glutamylcysteine synthetase.

Minor points:

Abstract

Two different fonts used, neither of which appear the same as the main body of the manuscript.

Introduction

The paragraph construction does not aid the flow of constructive thought. The short paragraphs are basically bullet points. The construction of the introduction needs revising to present aspects of each subsection together.

Adverbs and fronted adverbials have been employed liberally.  These should be avoided in scientific writing. For example, line 38: remove “indeed”; line 64: remove “in fact”; line 75: remove “therefore”, and so on.

Line 39: insert a space between sentences

Line 52: remove the extra space before “[4]”.

Line 69: the final sentence of the paragraph needs a reference

Results

Figure 1: the images do not clearly show what is stated in the text; check that all images are x20 magnification. Is there a way to quantify the claims made in the text? The figure legend needs to explain the images better to show the reader what they represent.

Figures 2 and 3: the authors state that the availability of data is not applicable. The availability of data is good practice and in this case essential for the reader to gauge reliability of the conclusions drawn from the figures presented.

Author Response

We wish to thank the Reviewer for her/his suggestions. We have included a quantification of expression levels of the master regulator of inflammation, TNFα to validate the data on the anti-inflammatory role of ranolazione. We have opted to measure mRNA, rather than protein levels, in order to employ a quantitative technique, such as Real-Time RT-PCR. The data obtained here confirm the results we published a couple of years ago demonstrating reduced IL-6 plasma levels in diabetic rats treated with ranolazine, as compared to the HFD/STZ vehicle group. 

Minor points:

  • We apologize for the use of different fonts. This has been modified.
  • We have extensively revised the Introduction section, according to the Reviewer suggestions. We have tried to reduce the inappropriate use of adverbs and will be happy to further modify the sentences that the Reviewer still feels may need improvement.
  • We have checked the images and we confirm that they are indeed all at 20x magnification. We will really appreciate if the Reviewer may specify which image s/he thinks is not of the required size to allow us to substitute it with a different one. We will also supply additional details on the quantification if needed.
  • We apologize for the misinterpretation of the “data availability” question. All data are available in our laboratory, and we have already supplied, as requested, all WB images.

Round 2

Reviewer 1 Report

The authors have addressed the initial concerns in the revised manuscript. In the revised manuscript in line 26 & 27, the authors state “we obtained additional support for the potential use of ranolazine to counteract T2DM- associated cognitive decline”. Do the authors intent to state that the results obtained in the present study supports the use of ranolazine to counteract T2DM- associated cognitive decline? If so, the sentence needs to be rephrased  by replacing the word “support” with the word “evidence”.

Author Response

We thank the Reviewer for her/his additional revision and have modified the final sentence of the abstract according to her/his suggestion

Reviewer 2 Report

Thank you for your response and revisions, which satisfactorily address my concerns. The only additional minor point is that streptozotocin is mis-spelled as streptozotocine in figure 4.

Author Response

We wish to thank the Reviewer for carefully reading the updated version of our manuscript. We have corrected the spelling of the word streptozotocin as suggested.